# Dynamic Walking of a Legged Robot in Underwater Environments

**DOI:** 10.3390/s19163588

**Published:** 2019-08-17

**Authors:** Gerardo Portilla, Roque Saltarén, Francisco Montero de Espinosa, Alejandro R. Barroso, Juan Cely, Oz Yakrangi

**Affiliations:** 1Universidad Politécnica de Madrid, Escuela Técnica Superior De Ingenieros Industriales, 28006 Madrid, Spain; 2Centro de Automática y Robótica, 28006 Madrid, Spain; 3Instituto de Tecnologías Físicas y de la Información “Leonardo Torres Quevedo”, 28006 Madrid, Spain

**Keywords:** underwater legged robot, dynamic walking, mechanical sensor

## Abstract

In this research, the dynamic walking of a legged robot in underwater environments is proposed. For this goal, the underwater zero moment point (Uzmp) is proposed in order to generate the trajectory of the centre of the mass of the robot. Also, the underwater zero moment point auxiliary (Uzmp aux.) is employed to stabilize the balance of the robot before it undergoes any external perturbations. The concept demonstration of a legged robot with hydraulic actuators is developed. Moreover, the control that was used is described and the hydrodynamic variables of the robot are determined. The results demonstrate the validity of the concepts that are proposed in this article, and the dynamic walking of the legged robot in an underwater environment is successfully demonstrated.

## 1. Introduction

As oceans occupy a huge part of the Earth [1], submarine robots are under constant development [2,3,4] for exploration, construction, mining and even aquaculture. However, the constant movement of the water causes instability in such robots. For this reason, legged robots are proposed as an alternative to achieve more stable exploration in an underwater environment.

To better understand the contribution of this article, Figure 1 shows the development of underwater locomotion, wherein the vertical axis indicates the type of locomotion (static walking, dynamic walking, jumping), and the horizontal axis shows the year of publication of these developments.

Concerning static walking, which is based on a centre of mass that is within the support polygon, you can see three important developments for underwater environments: (1) a six-legged underwater robot proposed by the researchers Junichi et al. [5]; (2) the Crabster CR200 robot, also with six legs, but with a more hydrodynamic structure [6,7,8]; and (3) a quadruped, humanoid robot proposed by researchers Roque Saltarén et al. [9].

On the concept of dynamic walking, there has been very little development. Yuta Kojio et al. [10] proposed the dynamic walking of a semi-submerged robot, as observed in (4). Being semi-submerged, the centre of mass is out of the water, so they did not make variations to the zero moment point. Moreover, hydrodynamic variables such as hydrodynamic mass are not considered. However, as noted in our investigation, this has great relevance in underwater dynamic walking.

In regard to jumping, the robot SILVER (5) was proposed by researchers Giacomo Picardi et al. [11,12]. This robot was designed with the aim of jumping in underwater environments. The interesting thing about their research is that they proposed the U-SLIP model (Underwater Spring Loaded Inverted Pendulum), a spring–mass model, which allows the development of jump control in underwater environments.

Noting the lack of development in the field of underwater walking dynamics, in this article the dynamic walking of an underwater legged robot is proposed. In order to reach this goal, a new concept is introduced: the underwater zero moment point (Uzmp). This is used to generate the centre of mass trajectory in such way that the robot can keep balance in the walking process. Additionally, knowing that underwater environments are constantly disturbed by the movement of water, the underwater zero moment point auxiliary (Uzmp aux.) is proposed. This is used to generate each step of the robot for stabilization in the event of water disturbance.

For the demonstration of the proposed concepts, a prototype of a planar bipedal robot was developed with a hydraulic actuator and a mechanical sensor that measures the velocity of the fluid relative to the centre of mass of the robot. The theoretical foundation and experimental validation of the sensor are found in [13,14].

## 2. Materials and Methods

### 2.1. Underwater Inverted Pendulum (UIP)

Figure 2 shows the Underwater Inverted Pendulum (UIP) model. It can be seen that the forces that affect a pendulum underwater are weight (mg), buoyancy (B), hydrodynamic mass (m’) and hydrodynamic damping (D).

#### 2.1.1. Simplification of Buoyant Force

The fluids generate a pushing force. The resultant force between the weight and that force can be resolved using [15]
(1)$$w_{u}=mg-\rho gV,$$
where $w_{u}$ is the resulting force due to buoyancy, m is the mass of the body, g is the gravity, ρ is the density of the fluid and V is the volume of the body. Clearing mass and gravity, we obtain
(2)$$w_{u}=mg\left(1-\left(\frac{\rho }{\rho_{i}}\right)\right),$$
where $\rho_{i}$ is the density of the body. We can see that the relationship between the densities is a constant. Hence, to simplify the Equation, it can be replaced by a constant λ in such a way that the resulting force is expressed as
(3)$$w_{u}=mg\lambda ,$$

#### 2.1.2. Hydrodynamic Damping

Hydrodynamic damping D(v) depends on two variables—linear damping and quadratic damping—as shown in the following equation [16,17].
(4)$$D\left(v\right)=X_{u}\dot{x}+X_{u\left|u\right|}\left|\dot{x}\right|\dot{x},$$
where $\dot{x}$ is the velocity of the centre of mass, $X_{u}$ is the constant of the linear damping, and $X_{u\left|u\right|}$ is the constant of quadratic damping.

Applying the sum of moments in the pendulum, we obtain the following.
(5)$$mg\lambda x+z\left(X_{u\left|u\right|}\left|\dot{x}\right|+X_{u}\right)\dot{x}-z\left(m^{\prime }+m\right)\ddot{x}-\tau =0,$$

Clearing $\ddot{x}$, we obtain the Underwater Inverted Pendulum model:(6)$$\ddot{x}=\frac{mg\lambda x}{z\left(m^{\prime }+m\right)}+\frac{\left(X_{u\left|u\right|}\left|\dot{x}\right|+X_{u}\right)\dot{x}}{\left(m^{\prime }+m\right)}-\frac{\tau }{z\left(m^{\prime }+m\right)},$$
where $\ddot{x}$ is the acceleration of the centre of mass, $\dot{x}$ is the velocity, and x is its position, m the mass of the pendulum, z is its height, $m^{\prime }$ is the hydrodynamic mass of the pendulum, $X_{u\left|u\right|}$ is the constant of the quadratic damping, and $X_{u}$ is the constant of the linear damping. Lastly, τ is the torque generated at the point of support.

### 2.2. Underwater Zero Moment Point

The torque generated in the Underwater Inverted Pendulum must be equal to the torque generated by the Uzmp location, as shown in Equation (7).
(7)$$\tau -\left(mg\lambda \right)U_{ZMP}=0,$$

Therefore, replacing Equation (7) in Equation (6), we obtain the underwater zero moment point as follows
(8)$$U_{ZMP}=x+\frac{z\;\left(X_{u\left|u\right|}\left|\dot{x}\right|+X_{u}\right)\dot{x}}{mg\lambda }-\frac{z\;\left(m+m^{\prime }\right)\ddot{x}}{mg\lambda },$$

#### Walking Pattern Generation

For the generation of the walking path, the Kajita method [18] with a model predictive control (MPC) should be used. To use this method, it is first necessary to linearize the $U_{ZMP}$ of Equation (8), obtaining the following.
(9)$$U_{ZMP}=x+\frac{z\;\left(2X_{u\left|u\right|}{\left|\dot{x}\right|}_{0}+X_{u}\right)\dot{x}}{mg\lambda }-\frac{z\;\left(m+m^{\prime }\right)\ddot{x}}{mg\lambda },$$
where ${\left|\dot{x}\right|}_{0}$ is the constant linearization velocity at which the centre of mass moves.

Let us define a new variable $u_{x}$ as the input of Equation (9),
(10)$$\frac{d\ddot{x}}{dt}=u_{x},$$
and the state space,
(11)$$\frac{d}{dt}\left[\begin{array}{c} x\\ \dot{x}\\ \ddot{x} \end{array}\right]=\left[\begin{array}{ccc} 0 & 1 & 0\\ 0 & 0 & 1\\ 0 & 0 & 0 \end{array}\right]\left[\begin{array}{c} x\\ \dot{x}\\ \ddot{x} \end{array}\right]+\left[\begin{array}{c} 0\\ 0\\ 1 \end{array}\right]u_{x}$$

For the output matrix, it is necessary to use Equation (9).
(12)$$\left[\begin{array}{c} U_{zmp}\\ x\\ \dot{x} \end{array}\right]=\left[\begin{array}{ccc} 1 & \frac{z\;\left(2X_{u\left|u\right|}{\left|\dot{x}\right|}_{0}+X_{u}\right)}{mg\lambda } & \frac{-z\;\left(m+m^{\prime }\right)}{mg\lambda }\\ 1 & 0 & 0\\ 0 & 1 & 0 \end{array}\right]\left[\begin{array}{c} x\\ \dot{x}\\ \ddot{x} \end{array}\right]$$

In this way, it is possible to obtain the output variables $U_{zmp}$, x, and $\dot{x}$. Finally, a predictive control was used, and the position of the centre of mass (*x*) and its velocity ($\dot{x}$) were obtained, as seen in Figure 3.

### 2.3. Reaction Step for Balance Recovery

Underwater environments are subject to constant disturbances, such as waves, currents, etc. For this reason, it is important to implement an underwater zero moment point auxiliary ($U_{ZMP}{}_{aux}$) so as to compensate the balance of the robot.

Figure 4 shows the inverted underwater pendulum, this time with the fluid in motion (*V*).

Figure 4 shows the $U_{zmp}$ displacement, due to the fluid disturbance, this variation is defined by the following Equation;
(13)$$\left(U_{{zmp}_{aux}}\right)mg\lambda =z\left(X_{u\left|u\right|}\left|v\right|+X_{u}\right)v$$

Clearing $U_{{zmp}_{aux}}$ (auxiliary underwater zero moment point), we obtain
(14)$$U_{zmp_{aux}}=\frac{z\left(X_{u\left|u\right|}\left|v\right|+X_{u}\right)v}{mg\lambda }$$
where v is the fluid velocity, which is measured by the mechanical sensor of the robot. When this variable is measured during the first 0.5 s, the maximum value is used to generate the reference.

This reference adds to the initial reference of the Uzmp of walking, and then the MPC is employed to generate the trajectory of the centre of mass (see Figure 5).

### 2.4. Dynamic and Control Position

#### 2.4.1. Inverse Dynamic

The inverse dynamics of the legged robot in an underwater environment [19] are defined by the following Equation.
(15)$$M\left(q\right)\ddot{q}+C\left(q,\dot{q}\right)\ddot{q}+D\left(q,\dot{q}\right)\dot{q}+g\left(q,B_{B}^{I}\right)=\tau -J^{T}f_{r},$$
where M(q) is the inertial matrix including the terms of hydrodynamic mass; $C\left(q,\dot{q}\right)$ is the Coriolis and centripetal matrix; $D\left(q,\dot{q}\right)$ is the hydrodynamic damping matrix; $g\left(q,B_{B}^{I}\right)$ is the vector of gravity and buoyancy; τ is the vector of torques and forces generated in each actuator; $\ddot{q}$, $\ddot{q}$, and q are the vectors of the accelerations, speeds, and positions of the actuators, respectively; $J^{T}$ is the Jacobian transpose of the leg; and $f_{r}$ is the vector of the reaction forces between the robot foot and the ground.

These reaction forces of the legged robot can be calculated by the following Equation [20].
(16)$$\left[\begin{array}{ccc} I & I & \ldots \\ r_{1}x & r_{2}x & \ldots \end{array}\begin{array}{c} I\\ r_{i}x \end{array}\right]\left[\begin{array}{c} F_{1}\\ F_{2}\\ \begin{array}{c} \vdots \\ F_{i} \end{array} \end{array}\right]=\left[\begin{array}{c} F_{com}\\ T_{com} \end{array}\right]$$
where $r_{i}$ is the vector between the CoM (centre of mass) and the foot *i*, *I* is the unitary matrix, $F_{i}$ is the forces vector of the reaction in the foot *i*, and $F_{com}$ and $T_{com}$ are the forces vectors and the torques, respectively, that are applied in the centre of mass of the robot. In the case of an underwater environment, these vectors are defined by the following Equation.
(17)$$\left[\begin{array}{c} F_{com}\\ T_{com} \end{array}\right]=\left[\begin{array}{c} \left(m+m^{\prime }\right)\vec{a}\\ I_{C}\vec{\alpha } \end{array}\right]+\left[\begin{array}{c} (X_{u\left|u\right|})\left|\vec{v}\right|\vec{v}\\ 0 \end{array}\right]+\left[\begin{array}{c} (X_{u})\;\vec{v}\\ 0 \end{array}\right]$$
where $\vec{a}={\left[\begin{array}{ccc} a_{x} & a_{y} & \lambda a_{z} \end{array}\right]}^{T}$ is the vector of the accelerations in the three axes of the centre of mass. Note that the third vector in the z-axis is multiplied by λ, which is the buoyancy factor shown in Equation (3). $I_{C}$ is the inertial matrix of the robot body and $\vec{\alpha }$ is the vector of angular accelerations in the three axes. $\vec{v}$ is the vector of the relative velocity in the three axes of the fluid with the trunk of the robot; this value is calculated by the robot’s mechanical sensor.

Depending on the number of feet in contact with the ground, Equation (16) may or may not have multiple solutions. To solve this problem, the criterion of the friction force is used, in which forces must satisfy the following Equation.
(18)$$\sqrt{f_{x}+f_{y}}\leq \mu f_{z},$$
where $f_{x}$, $f_{y}$, and $f_{z}$ are the reaction forces in each foot and μ is the coefficient of the static friction of the ground.

#### 2.4.2. Control Position

The control strategy that was used in the robot is a model of dynamic compensation control. As shown in Figure 6, a proportional–integral controller (PI) [21] was used. It is compensated [22,23] by the torques, which are calculated by the inverse dynamics. This value is multiplied by the value of *Kp*, which is easily adjustable in the robot. This method allows the control to be more adaptive, since during the period in which the foot of the robot is on the ground, the torques and forces increase drastically. This method allows such variability to be compensated during the calculation of the inverse dynamics.

### 2.5. Description of Prototype

For the experimental test, a planar bipedal robot was built. The advantage of using this type of robot is that it allows you to test two-dimensional models [24,25,26,27], such as $U_{ZMP}$. Figure 7 shows some planar bipedal robots.

Figure 8 shows the degrees of freedom of our underwater planar bipedal robot. As can be seen, it has two legs (leg1, leg2) with two degrees of freedom, one rotational (Q1, Q2), and another linear (L1, L2). This is tied to an axis attached to a base through a universal joint, allowing the robot to have two degrees of spatial freedom.

Figure 9 shows a biped robot with four hydraulic actuators, which work at a hydraulic pressure of 20 bars. This robot has a height of 90 cm, a width of 22.2 cm and weighs 4 kg. The mechanical sensor was developed by our research group, and it is made to measure the fluid velocity in all three directions. The robot has three inertial measurement unit (IMU) sensors; one in the centre of mass in order to calculate the linear accelerations, angular positions and speeds, and another in each leg in order to calculate the angular position. Moreover, it has two flexible resistive sensors; by their deformation it is possible to measure the displacement of the linear actuators in the legs. Also, there are two contact sensors in each foot. This type of sensor was chosen because it is waterproof, as each of them is covered with resin.

Figure 10 shows the mechanism that was used to transform the linear movement of the actuators into a rotational movement to rotate the leg. The mechanism is based on a sliding mechanism perpendicular to the line actuator; when the cylinder bar is introduced, this rotates the leg clockwise (see Figure 10a), and when it is it comes out, the leg rotates counterclockwise (see Figure 10b). This mechanism of action is applied to each leg (see Figure 10c).

The mechanical sensor measures the velocity and direction of the fluid through the parallel mechanism [28,29,30,31] (see Figure 11), which uses passive actuators (springs). The platform of the mechanism is connected to a spherical body. Within this body, there is an inertial measurement unit (IMU). When the fluid collides with the sphere, the parallel mechanism is deformed, from which it is possible to determine the angles of the platform and the acceleration of the sphere by using the IMU. Using inverse kinematics, the deformation of the springs is calculated. From the deformation of the springs, the force of each actuator can be determined. Then, using dynamics, the drag force in the sphere is calculated. Finally, the velocity of the fluid is calculated by the hydrodynamic Equation of the drag force. The theoretical explanation of this sensor is shown in the paper by the authors of [13] and the demonstration experiment is shown in the paper by the authors of [14].

The robot control software was developed in LabVIEW and the hardware architecture is shown in Figure 12. Using an Arduino DUE, the data of the sensors were extracted and transferred to a computer by a USB cable. In this way, high-level control is executed and the positions of each actuator are sent to the C-Rio, where the low-level control described in Section 2.4.2 is executed. Also, it generates the voltage signals, which are sent to the current converter, and activates the servo valves.

Figure 13 shows the submerged robot in an underwater environment at a depth of 4 m. It can be observed that this is supported by the axis that is connected by a universal connection to the metal base. In addition, it is possible to see the hydraulic hoses and the Arduino DUE, which is covered with resin.

#### 2.5.1. Hydrodynamic Mass

Because the computational methods for calculating hydrodynamic mass are still in development, the method that was used to determine the hydrodynamic mass is the Small Oscillation Method, by means of mass-spring free vibration [32]. This is an experimental method which has been shown to have good results with scale models [33,34,35].

Figure 14 shows the experiment. The robot’s hip model at 0.5 scale can be seen submerged in water. This model and a weight of 0.5 kg are suspended by a spring, while a linear sensor measures the displacement of vibration.

The hydrodynamic mass of a body is determined by the following Equation [36].
(19)$$m^{\prime }=C_{m}\rho_{w}V$$
where $C_{m}$ is the hydrodynamics mass coefficient, $\rho_{w}$ is the density of the water and V is the volume of the body.

The Equation of motion is determined as follows [37]
(20)$$y=A_{Y}e^{-\zeta w_{d}t}\cos\left(w_{d}t\right)$$
where y is the amplitude, $A_{Y}$ is the maximum amplitude, ζ is the damping factor, $w_{d}$ is the oscillation frequency and t is the time. The oscillation frequency is determined by the following Equation [37].
(21)$$w_{d}=w_{n}\sqrt{1-\zeta^{2}},$$
where $w_{n}$ is the natural frequency without damping. Since ζ is normally small compared with unity, the damped natural angular frequency, $w_{d}$, can be approximated to $w_{n}$, the undamped natural angular frequency [37],
(22)$$w_{d}=w_{n}\sqrt{1-\zeta^{2}}≅w_{n}$$
where [36],
(23)$$w_{n}=\sqrt{\frac{k}{m+m'}}$$
where *K* is the stiffness of the spring (120 N/m), *m’* is the hydrodynamic mass, and *m* is the mass of the model with the weight addition of 0.5 kg. By using a linear encoder, the displacements can be measured and the natural frequency can be calculated. In this way, it is possible to determine the hydrodynamic mass of the model and then to calculate $C_{m}$ by Equation (19).

Figure 15 shows a free fall experiment, where the amplitude decays exponentially. By equating the amplitude Equation shown in the figure with Equation (26), the values ζ=0.0339 and $w_{d}=4.36$ can be obtained. The damping factor value was observed to be quite low, indicating that the vibration frequency can approximate very well to the natural frequency.

Figure 16 shows the results of $C_{m}$ that were obtained in 10 experiments. The mean value of $C_{m}$ was found to be 1.89. This value is replaced in Equation (19) with the real volume of the robot to obtain the hydrodynamic mass of 4.2 kg.

To verify that our experimental results are correct, we approximated the geometry of the robot’s hip to a parallelepiped. Figure 17 shows the hydrodynamic mass coefficient for a parallelepiped that is in motion on the x-axis [32]. At the right-hand side, the dimensions of the robot’s hip are shown, approximating it to a parallelepiped. Using the table with the given dimensions, we obtained $C_{m}\approx 1.5$, which is very close to the value we found through the experiment ($C_{m}=1.89$), being reasonably higher since the geometry is more complex. Therefore, we can ensure that the value of $C_{m}=1.89$ found by the experiment is correct.

In order to determine the hydrodynamic mass of the legs and axis, the geometry that was assumed is the geometry of the cylinder; the hydrodynamic mass coefficient of the cylinder is $C_{m}=1$ [36]. Table 1 shows the mass and the hydrodynamic mass of each body.

In Table 1, it is possible to see that the total mass of the hip represents 77.7% of the total mass of the robot. Taking into account that the hip has the highest accelerations, for our $U_{zmp}$ model only the value of the hip is considered.

#### 2.5.2. Damping Coefficients

The hydrodynamic damping coefficients [38,39] of the robot were obtained by CFD (computational fluid dynamics) with a series of experiments by varying the fluid speed, as shown in Figure 18. First, the applied hydrodynamic forces in the body of the robot are obtained. By performing a regression, the quadratic Equation is obtained, from which it is possible to deduce the quadratic coefficient $X_{u\left|u\right|}=134.4\;{\mathrm{Ns}}^{2}/m^{2}$ and linear coefficient $X_{u}=18.6\;\mathrm{Ns}/m$.

### 2.6. Experiment

Two experiments were carried out; walking for a period of 54 s and a reaction step for balance recovery. In last experiment, two perturbations in the fluid were generated by a square aluminium plate of 30 cm. The constants used in the experiments are shown in Table 2.

## 3. Results

### 3.1. Walking

Figure 19 shows a series of photographs of the walking period during 54 s (see Appendix A).

Figure 20 shows the trajectory of the CoM (red line) that is generated by the MPC and Uzmp. The feet trajectory of the robot is also shown along the x-axis and the z-axis.

Figure 21 shows the position control results of each actuator, where the blue line is the reference position and the orange line is the position obtained by the actuator.

Figure 22 shows the results of the velocities in the three directions that were sensed by the mechanical sensor. It can be seen that the velocity along the x-axis maintains an average of 0.02 m/s. The y-axis velocity has an average of 0.003 m/s, much smaller than the velocity along the x-axis. However, it is greater than the velocity along the z-axis, due to the fact that the robot is walking in a circular manner. So even if the sensor moves with the robot, there is a y-direction component due to the centrifugal force. The speed along the z-axis is very small since the robot does not move or at least avoids moving in that direction. Hence, these velocity measurements are a product of disturbances in the environment that are caused by walking.

### 3.2. Reaction Step for Balance Recovery

Figure 23 shows a photographic series of the experimentation process for the reaction of the step to recover the balance of the robot (see Appendix A). In the first photograph, the aluminium plate is observed in the yellow circle to generate the disturbance and the yellow arrow indicates its movement. The blu, e arrow indicates the movement of the fluid that is generated to disturb the robot. Two disturbances were made; the first a time of 0 s reaches stability at 4 s. Then another disturbance is generated at 5 s and the robot regains its stability at 7 s.

Figure 24 shows the velocity that is measured by the mechanical sensor. Two disturbances can be clearly seen—the first generated a velocity along the x-axis of up to −0.15 m/s (the negative value is because the sensor also measures the direction of the fluid), and the second disturbance caused the fluid velocity to reach an average of −0.2 m/s. The disturbances were also generated along the y- and z-axes.

Figure 25 shows the Uzmp aux (blue line), which is calculated by Equation (14). The orange line is the reference generated with the maximum value of the Uzmp aux; at the beginning a maximum reference value of −0.05 m was generated and in the second case a value of −0.1 m was obtained.

Once the reference of the Uzmp is defined, the MPC can be used to generate the trajectory of the centre of mass, as shown in Figure 26. In this case, the value is negative, so it is understood that the robot steps back in order to recover its balance.

Figure 27 shows the reference positions of each actuator (blue line) at the moment of the reaction, obtained from the inverse kinematics. Also, from the positions of the actuators (orange line), it can be seen that in the case of Q1 and L1, the first one has a larger error. This situation is normal because at that moment the disturbance is taking place.

## 4. Discussion

Figure 20 shows the path generated by the Uzmp. First, a low speed is generated for walking, approximately 0.02 m/s, as shown by the measurement of the mechanical sensor. There are two main reasons why a low speed is obtained: First, as seen in the Uzmp Equation, is that the inertial force increases due to the hydrodynamic mass acquired by the robot, which is quite considerable as demonstrated in the experiment. On the other hand, the gravitational force decreases due to the buoyancy of the body; this makes the pendulum escape velocity very low, decreasing the maximum walking speed, and causes a transition to trotting or jumping. This analysis of a robot with terrestrial legs was performed with the Froude number. According to Alexander’s research [40] on walking bipeds and quadrupeds, walking gaits are abandoned at Froude numbers greater than about 0.5. Therefore, the same analysis should be done by proposing a Froude number for underwater environments. This would enable a better understanding of the maximum walking and transition speeds, which depend on hydrodynamic constants such as the hydrodynamic mass and buoyancy.

The walking speed is also limited due to the reaction force in the plane. Due to the buoyancy, the force in the z-direction of the foot decreases, reducing the limit of the reaction force. To prevent the robot from sliding, this reaction force in the plane must be greater than the force generated by the damping of the water, which depends on the hydrodynamic constants and the speed of the robot. Equating Equation (18) with Equation (4), we obtain the following.
(24)$$X_{u}\dot{x}+X_{u\left|u\right|}\left|\dot{x}\right|\dot{x}\leq \mu \;\left(mg\lambda \right).$$

Solving this inequality in the case of our planar robot, the centre of mass velocity must be $\dot{x}\leq 0.3\;m/s$. At velocities greater than this speed, the robot would slide. Obviously, this is a theoretical value, but it is still very low. In the real experiment, at the speed of 0.02 m/s, the robot did not present a slide. Upon increasing the speed, the robot could walk but presented a slide in certain areas. In order to overcome this speed limit, the hydrodynamic constants would have to be reduced, which could be achieved more hydrodynamic designs.

Regarding the proposal of Uzmp for walking underwater, researchers Yuta Kojio et al. [10] presented a proposal for a humanoid robot to walk semi-submerged. They proposed using the terrestrial ZMP plus an auxiliary variable, which only depends on the quadratic damping, without taking into account the hydrodynamic mass. However, we determined that this variable cannot be neglected, since its value is important; in our case, it causes the mass of the robot to double. On the other hand, not taking into account the linear damping at low speeds (less than 1 m/s) could also lead to error since, as Fossen [39] mentioned, linear damping at low speeds may be greater than quadratic damping. For this reason, both should always be taken into account.

The auxiliary Uzmp managed to generate a reference to absorb the hydrodynamic force of the fluid, as observed in Figure 25, causing the robot to step back in order to recover balance. There are different proposals for the absorption of external forces for terrestrial robots; these methods are mainly based on measuring the acceleration of the centre of mass of the robot. This makes sense in terrestrial environments, since it is very difficult to estimate the force of the disturbance before it affects the body. However, with our method we can estimate the force by measuring the speed of the fluid, and that can be done with either our proposed sensor or other more advanced sensors which could even estimate the speed of the fluid before it reaches the robot, such as Doppler effect sensors. This would allow the generation of the reference step before the water disturbance affected the robot.

Figure 21 and Figure 27 demonstrate that the control strategy applied in this investigation works. Compensation control is nothing new, but the fact that it is feedback based on the fluid velocity contributes a lot. The current robots with underwater legs do not measure the speed of the fluid, but rather assume at all times that the fluid is not in motion. This only has a limited range of application, as we know that fluid is always in motion, and that the robot is sensitive to small disturbances. Thus, it is necessary to know the relative velocity of the fluid surrounding the body; this is why the development of the mechanical sensor for the measurement of fluid velocity was important.

## 5. Conclusions

It can be concluded that the Uzmp proposed in this article gives satisfactory results, as it enables the robot to remain stable during walking. Furthermore, the Uzmp aux. was shown to be able to calculate the stabilization point of the robot in the face of disturbances, generating the trajectory of the centre of mass to react and not to fall. The mechanical sensor developed by our research group was also shown to be of great importance, both for the Uzmp and for the control of the robot. As seen in the Equations, the relative speed of the fluid surrounding the robot is an important variable to measure in order to achieve robot locomotion in underwater environments.

For future research, an underwater inverted pendulum model with thrusters could be proposed, so that the robot’s speed is not limited by the reaction force, compensating for either the hydrodynamic damping force or the buoyancy. It is very likely that future robots with underwater legs will have thrusters for navigation, though they can also be used for locomotion.

## Figures and Tables

**Figure 1 sensors-19-03588-f001:**
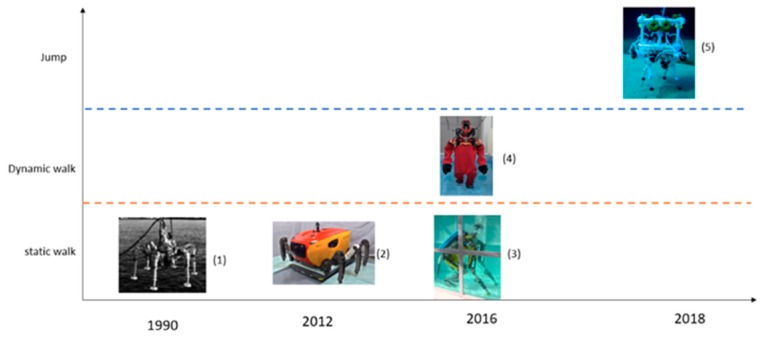
Development of underwater locomotion, where the locomotion types are represented on the vertical axis, and the year in which they were developed on the horizontal axis.

**Figure 2 sensors-19-03588-f002:**
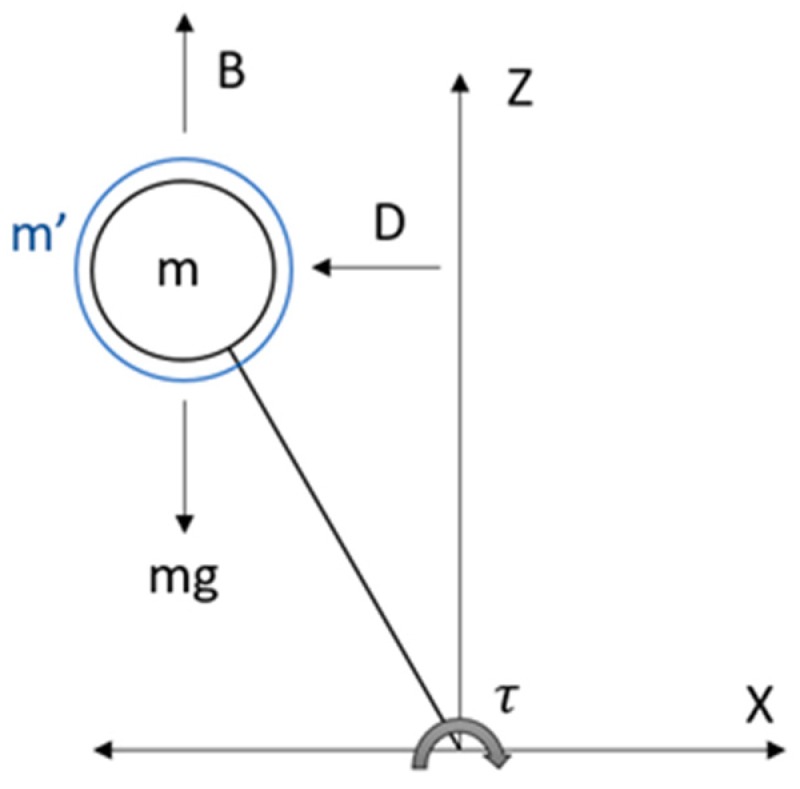
Inverted pendulum model in an underwater environment. Here, m’ is the hydrodynamic mass of the water, m is the mass of the robot trunk, D is the hydraulic damping force, mg is the weight, B is the buoyancy of the body and τ is the resulting torque.

**Figure 3 sensors-19-03588-f003:**
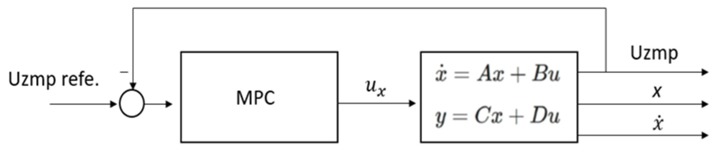
Model Predictive control for the calculation of the position and velocity of the centre of mass.

**Figure 4 sensors-19-03588-f004:**
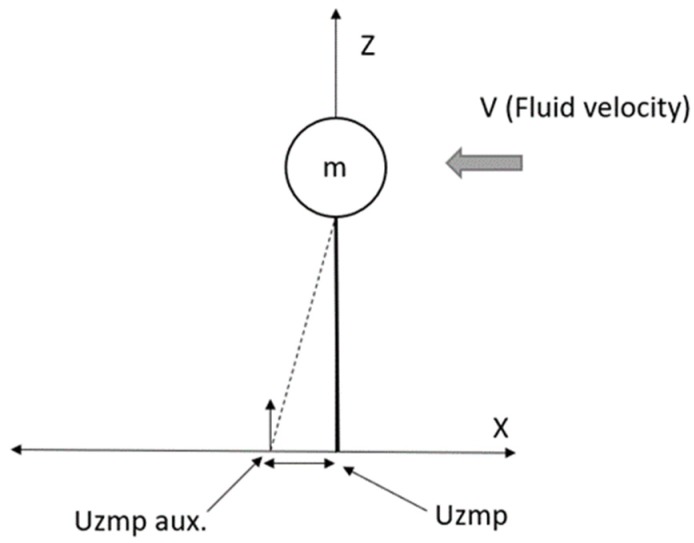
Displacement of the Uzmp when the inverted pendulum is disturbed by the movement of the fluid (V).

**Figure 5 sensors-19-03588-f005:**
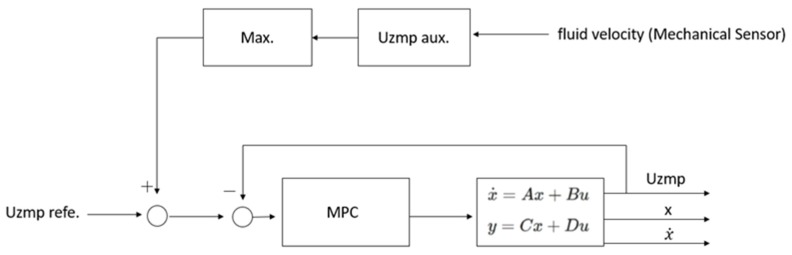
Reaction model step to recover balance.

**Figure 6 sensors-19-03588-f006:**
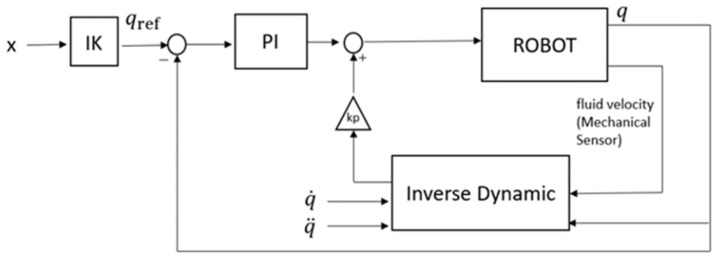
Position control by dynamic compensation.

**Figure 7 sensors-19-03588-f007:**
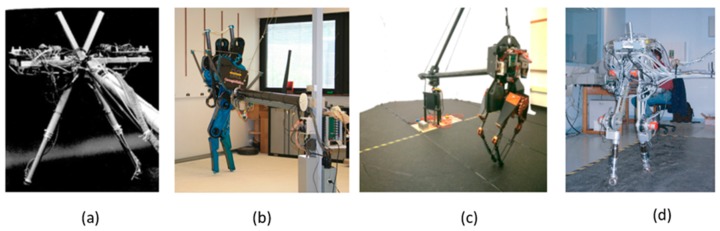
(**a**) A biped robot used by MIT’s leg laboratory, (**b**) the Mabel robot of the University of Michigan, (**c**) Atrias of the Oregon State University’s Dynamic Robotics Laboratory and (**d**) Rabbit of Laboratoire de Grenoble Automatique.

**Figure 8 sensors-19-03588-f008:**
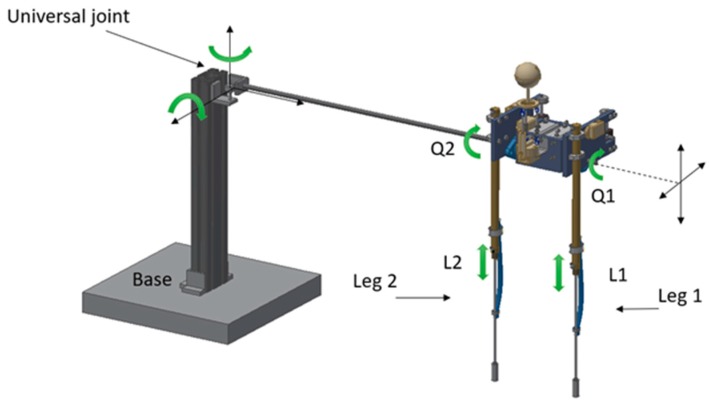
Planar bipedal robot for underwater walking.

**Figure 9 sensors-19-03588-f009:**
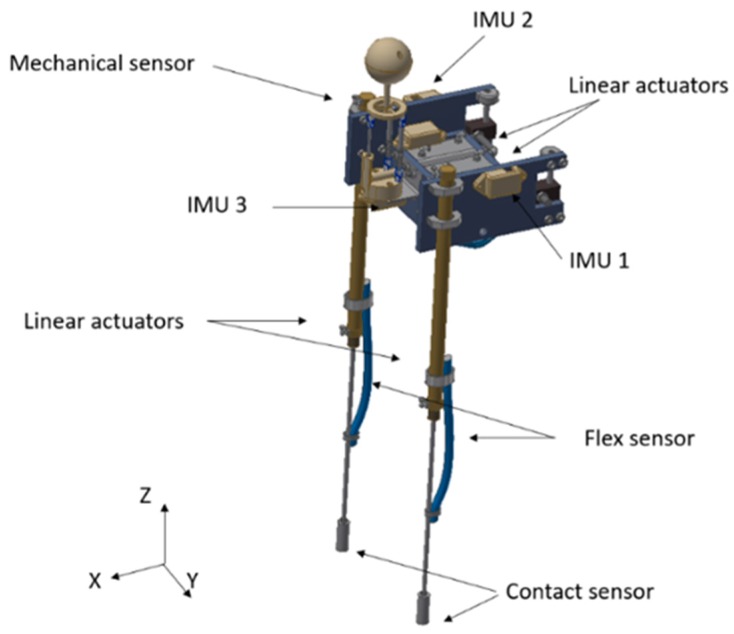
Bipedal robot sensors and actuators.

**Figure 10 sensors-19-03588-f010:**
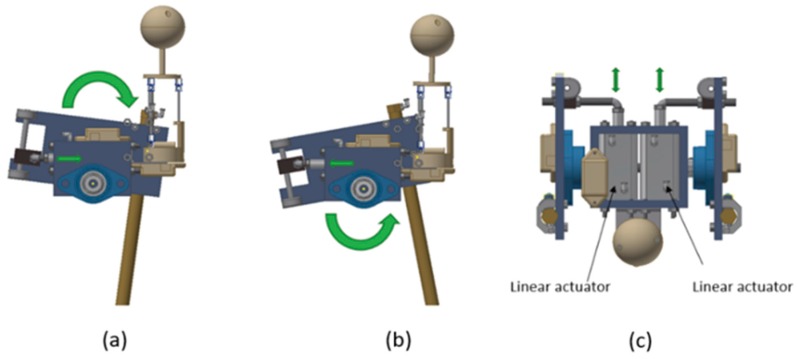
Leg rotation mechanism, (**a**) cylinder bar introduced, this rotates the leg clockwise, (**b**) cylinder bar outside, this rotates the leg counterclockwise, (**c**) top view of each actuator.

**Figure 11 sensors-19-03588-f011:**
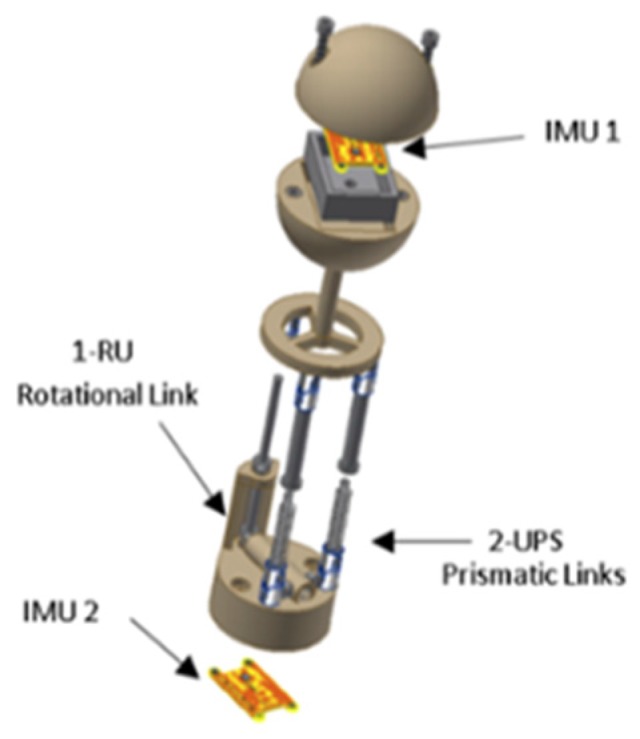
CAD design of the sensor. Inertial measurement unit 1 (IMU1) is in the sphere of the sensor, and IMU2 is in the base of the sensor. Also, the 2-UPS prismatic links and 1-RU rotational link are shown.

**Figure 12 sensors-19-03588-f012:**
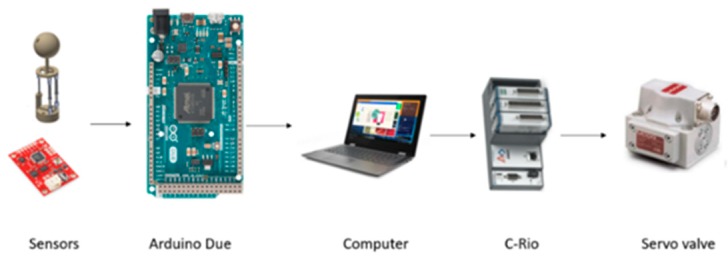
Hardware architecture of the legged robot.

**Figure 13 sensors-19-03588-f013:**
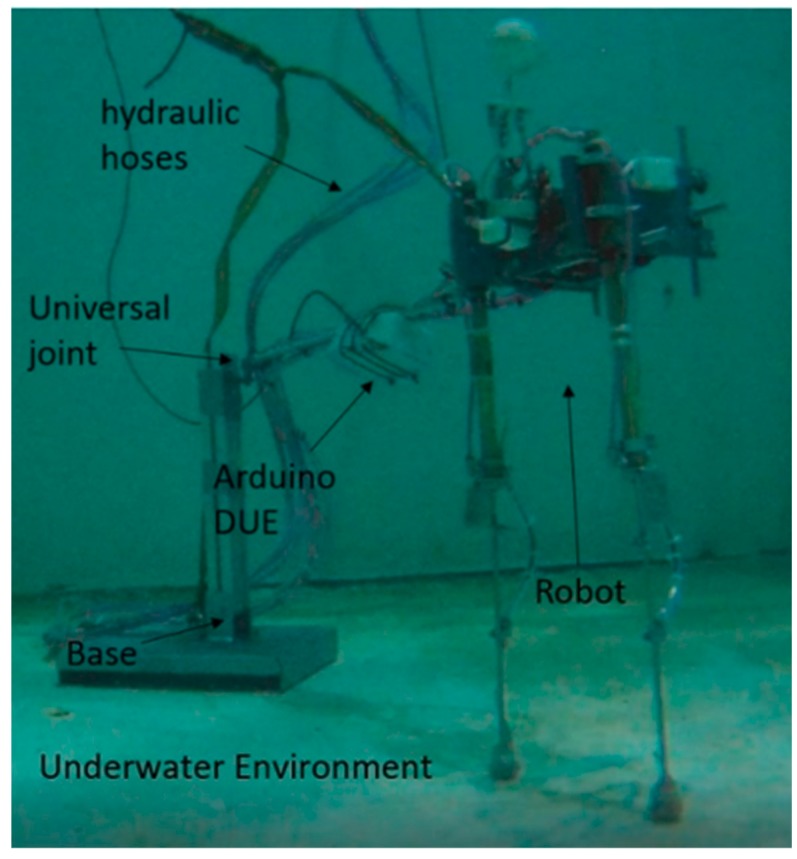
Legged robot in an underwater environment at a depth of 4 m.

**Figure 14 sensors-19-03588-f014:**
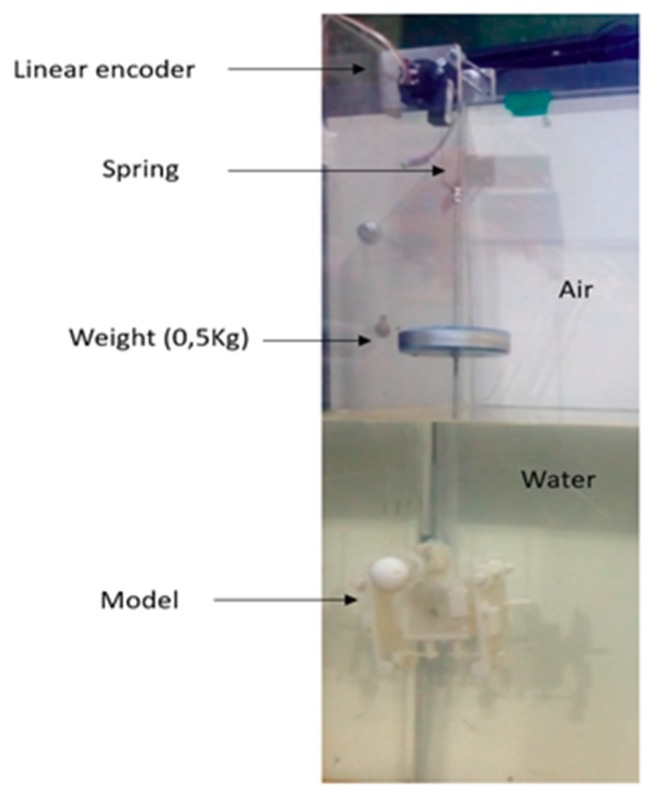
Experiment model for the determination of hydrodynamic mass.

**Figure 15 sensors-19-03588-f015:**
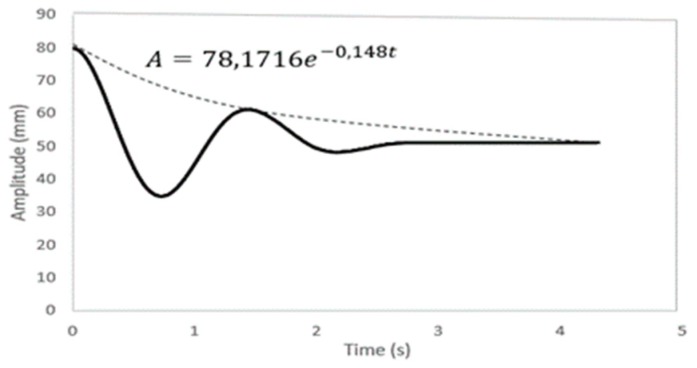
Oscillating movement of the free fall experiment.

**Figure 16 sensors-19-03588-f016:**
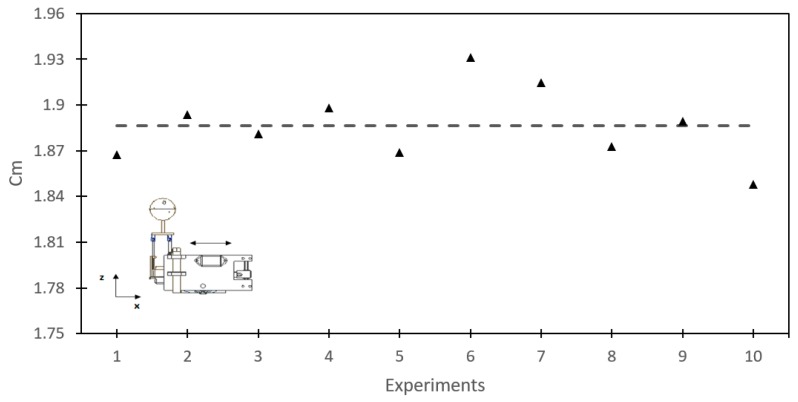
Hydrodynamic mass coefficients obtained in each experiment.

**Figure 17 sensors-19-03588-f017:**
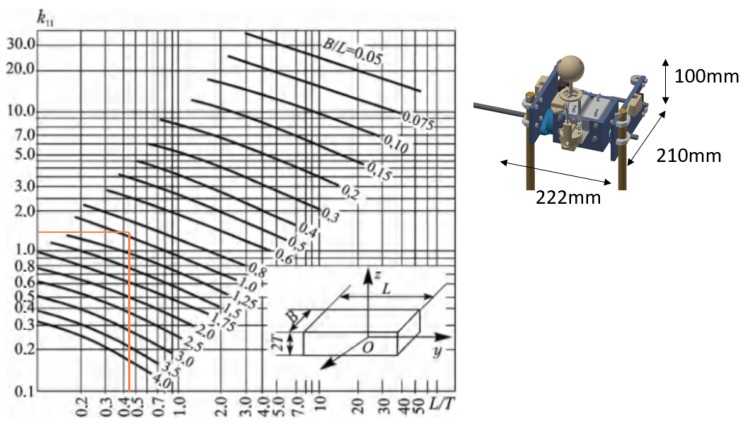
Hydrodynamic mass of a parallelepiped moving on the x-axis, where $k_{11}=C_{m}$.

**Figure 18 sensors-19-03588-f018:**
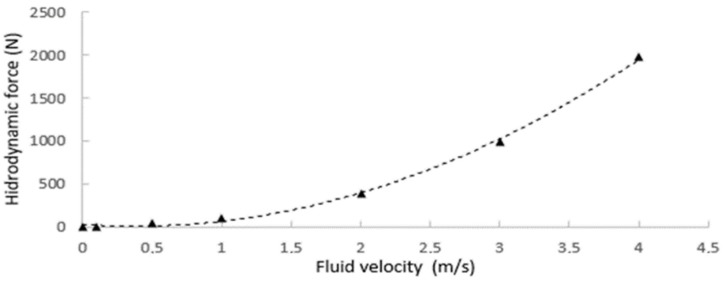
Hydrodynamic forces that are obtained by each fluid velocity to determine the damping constants.

**Figure 19 sensors-19-03588-f019:**
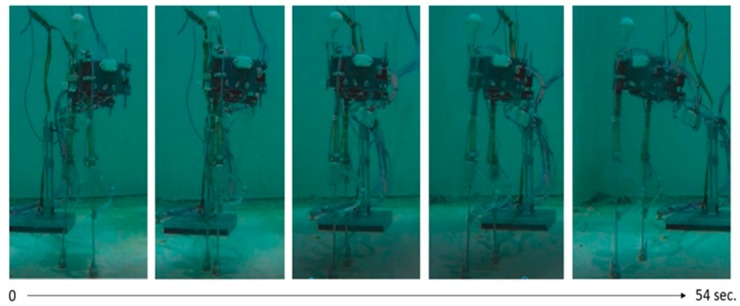
Photographic series of the walking process.

**Figure 20 sensors-19-03588-f020:**
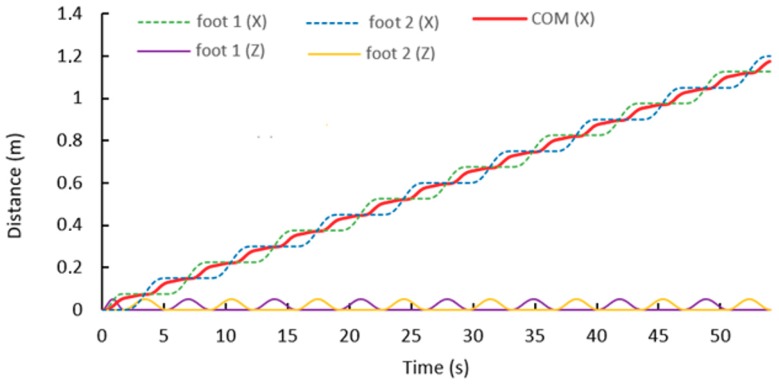
Trajectory of the centre of mass generated by the Uzmp and MPC.

**Figure 21 sensors-19-03588-f021:**
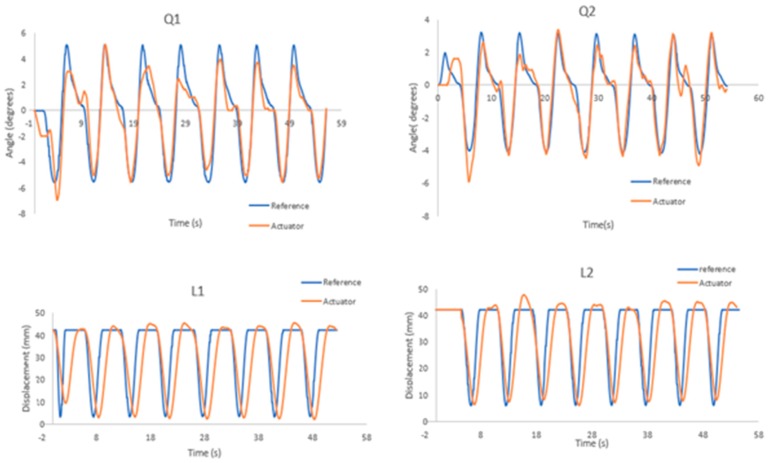
Position actuators in the walking process.

**Figure 22 sensors-19-03588-f022:**
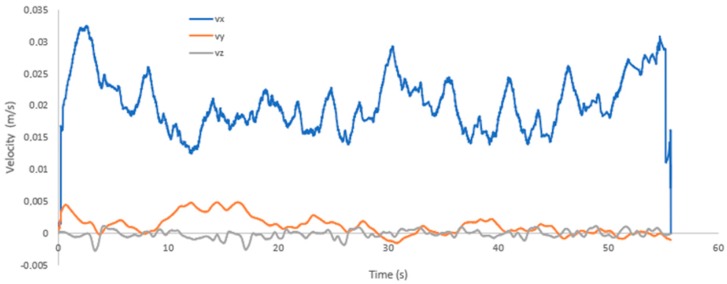
Speeds measured by the mechanical sensor in the walking process.

**Figure 23 sensors-19-03588-f023:**
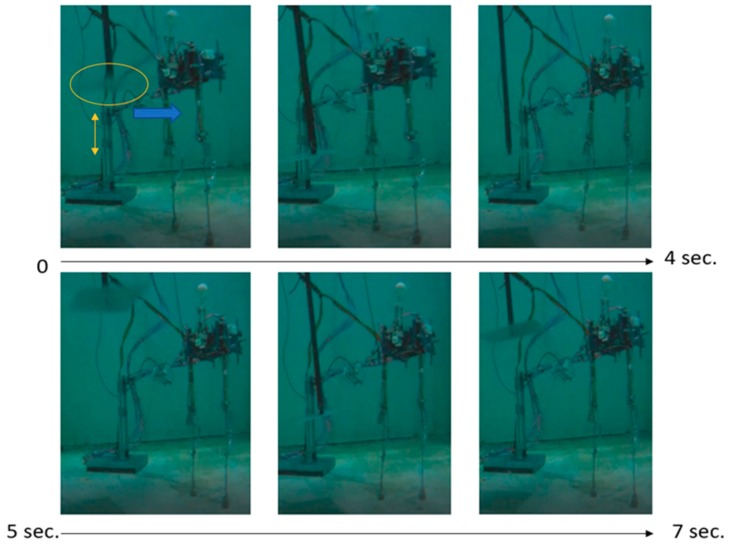
Photographic series of the reaction step experiment for balance recovery.

**Figure 24 sensors-19-03588-f024:**
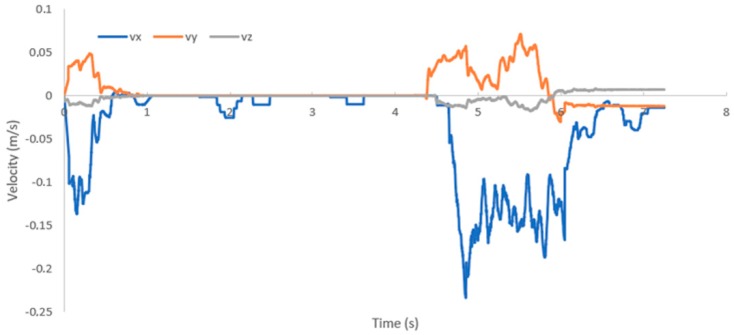
Fluid velocities that are measured by the mechanical sensor at the time of water disturbance.

**Figure 25 sensors-19-03588-f025:**
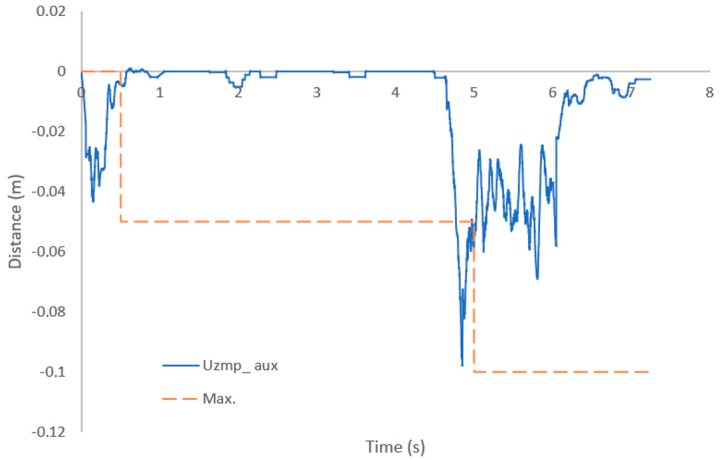
Underwater zero moment point auxiliar (line blue) and reference (orange line).

**Figure 26 sensors-19-03588-f026:**
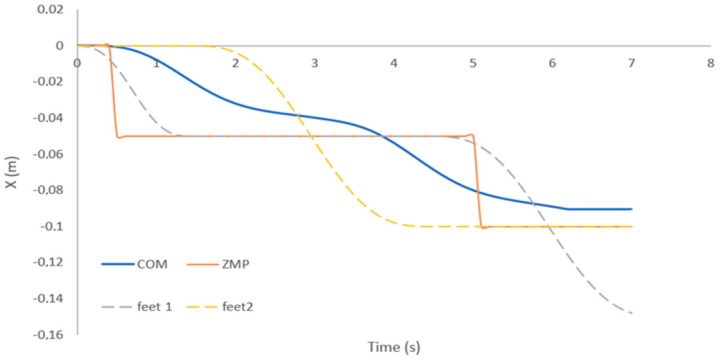
Trajectory of the centre of mass, generated by the MPC, in reaction to a disturbance.

**Figure 27 sensors-19-03588-f027:**
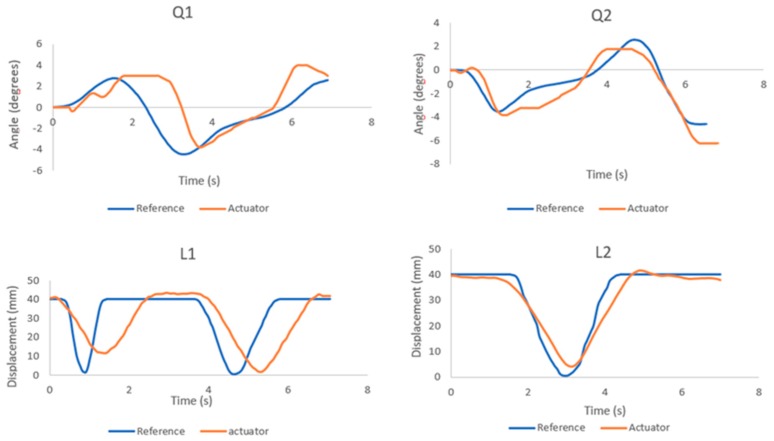
Position actuators in the balance recovery process.

**Table 1 sensors-19-03588-t001:** Added mass of each body.

body	m (kg)	m’ (kg)	m + m’ (kg)
Hip	4	4.2	8.2
Leg1	0.7	0.186	0.886
Leg2	0.7	0.186	0.886
axes	0.5	0.079	0.579
		sum	10.551

**Table 2 sensors-19-03588-t002:** Constants used in the experiment.

Constants	
*z*	0.7 m
$X_{u\left|u\right|}$	$134.4{\;\mathrm{Ns}}^{2}/m^{2}$
$X_{u}$	18.6 Ns/m
m	4 kg
$m^{\prime }$	4.2 kg
*g*	9.8 m/s^2^
λ	0.7
*MPC*	Prediction control = 90, Control horizon = 20
${\left|\dot{x}\right|}_{0}$	0.02 m/s
μ	0.8
*PI*	Q1 (0.04, 300); Q2 (0.04, 300); L1 (0.05, 150); L2 (0.05, 150)
*kp*	Q1 (−0.15); Q2 (−0.15); L1 (−0.1); L2 (−0.1)
*K*	120 N/m

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
