# Peer review of "Dynamic Walking of a Legged Robot in Underwater Environments"

_sensors, 2019, doi:10.3390/s19163588_

Round 1

Reviewer 1 Report

General comment:

The authors presented an extension of the renown ZMP for terrestrial walking. Alongside with a new model, the authors also developed a working (and walking) underwater robot, which is a remarkable technical achievement. It can be appreciated the effort in creating a functional prototype.

However, the paper suffers severe weaknesses under the presentation point of view, as well as under the methodological one. English is very poor, with an incredibly high number of typos. The language maybe influenced also the presentation of the “material and methods” section, which should be properly corrected accordingly to the comments below. Several points need to be clarified, including some part of the results, and additional tests are required to sustain the author claims.

Detailed comments:

M1. In the seminal paper on preview control of ZMP [Kajita et al. 2003] (which the authors correctly cite), the zero moment point equation is properly derived. The same is required also from the authors, since their model is significantly different from the one proposed in [Kajita et al. 2003], otherwise it will be very difficult for readers to follow their argumentations. Same considerations for the UZMP_aux.

M2. Moreover, in Eq. (3), X_u_u is the aggregate coefficient of the quadratic hydrodynamic drag, but the subsequent dependence on speed is only linear. Without the proper derivation of the equation, it is difficult to tell if it is correct or not: by trying to derive the equation myself, it appears to me that it should be X_u_u times dot{x} times absolute dot{x} (the correct dependence for quadratic speed). I suppose the same error is in Eq.s (5) and (6). On the other hand, Eq. (9) correctly reports the quadratic dependence X_v_v times {v} times absolute value of {v}.

M3. It would be also beneficial to have additional details on the mechanical structure of the robot. The Fig.8 is not useful, but additional information on the mechanism of robot's body and leg could allow to better appreciate the work of the authors, and as well understand the working mechanism of the underwater structure. In the present form of the paper, it appears that joint Q1 is not back driveable: if so and the robot is fixed to the underwater structure, it is difficult to understand how the robot can fall.

M4. Calculation of the added mass (which authors refer to as hydrodynamic mass) it is not so straightforward for the complex shape of their robot. It has to be clarified how they calculate it, especially since they used:

1.      Not damped oscillations hypothesis

2.      Neglectable contributions of legs and underwater structures

And others additional, minor hypothesis.I really think that their test, and the actual robotic system, will have very different values.

R1. Results mainly confirm that the control is able to follow the reference signal on robot joints, but they are not presented in a sufficient convincing way to tell that the developed UZMP actually works. Additional test, with more systematic comparisons have to be performed. Eventually, the comparison between the video (additional material) and the Fig. 16 is really puzzling me: the robot’s body clearly tilts backward several times, which should be clear from a negative forward velocity. As presented in Fig. 16, the forward speed is always positive, indicating that the robot’s body should never move backward.  

D1. The discussion is very poor. The authors should first point out the difference between previous models, why they needed to improve the existing literature, and finally prove that their new approach/model is actually describing the physical phenomenon. So far, it seems more a technical report than a scientific contribution.

Final comment:

Overall, I think that the authors are scratching the surface of a very interesting topic, with potential application in several fields (from underwater robotics, to space exploration and rehabilitation medicine). However, the paper need to be significantly improved with additional work, since both the methods that the results are too preliminary.

Reviewer 2 Report

The papar presented an underwater zero moment point method for the walking of an underwater robot with external perturbations. Experiments were conducted to demonstrate the effectness the concepts.

ZMP method is very commonly used in the legged robot. So the only novelty is the compensation for the balance of the robot. The input of this compensation module is the fluid velocity measured by mechnical sensor. It is a relative velocity between fluid and robot. Then, how does the proposed method work when actual fluid velocity equals to robot velocity (meased fluid velocity =0)?

In the video, the robot is walking very slowly. Can the authors explain why? We know the fluid is changing very fast under the water, so fast reaction is needed.

The introduction should be rewriten and the stat of art of control method for underwater robot against the wave and fluid should be added.

In the experiments, comparison with other method should be added to show the merits of the proposed method.

The discussion part looks more like “the conclusion”, and this should be modified.

The English of the paper should be improved. Some sentences are difficult to understand.

There’s no explaination about Q1, Q2, L1 and L2.

Round 2

Reviewer 1 Report

No additional comments.

Reviewer 2 Report

Authors have addressed my comments for this reason I recommend the publication of this paper